# Novel Therapies and Strategies to Overcome Resistance to Anti-HER2-Targeted Drugs

**DOI:** 10.3390/cancers14184543

**Published:** 2022-09-19

**Authors:** Manuel Gámez-Chiachio, David Sarrió, Gema Moreno-Bueno

**Affiliations:** 1Biochemistry Department, Medicine Faculty, Universidad Autónoma Madrid-CSIC, IdiPaz, 28029 Madrid, Spain; 2Centro de Investigación Biomédica en Red-Oncología (CIBERONC), 28029 Madrid, Spain; 3MD Anderson International Foundation, 28033 Madrid, Spain

**Keywords:** HER2 breast cancer, resistance, antiHER2 therapies

## Abstract

**Simple Summary:**

Drug resistance is the “Achilles’ heel” in current oncology. In this sense, the clinical management of HER2 breast carcinomas (tumors with overexpression/amplification of the ErbB2/HER2 oncogene) is still a challenge. Although a variety of anti-HER2 therapies (anti-HER2 antibodies, antibody–drug conjugates, and tyrosine kinase inhibitors) are available for these patients, the frequent appearance of innate and acquired tumor resistance to these treatments creates an urgent need to develop new and more effective therapeutic approaches. In this review, we discuss the most relevant clinical and pre-clinical advances in therapeutic strategies aimed at overcoming drug resistance in HER2 breast cancer patients in different stages of the disease.

**Abstract:**

The prognosis and quality of life of HER2 breast cancer patients have significantly improved due to the crucial clinical benefit of various anti-HER2 targeted therapies. However, HER2 tumors can possess or develop several resistance mechanisms to these treatments, thus leaving patients with a limited set of additional therapeutic options. Fortunately, to overcome this problem, in recent years, multiple different and complementary approaches have been developed (such as antibody–drug conjugates (ADCs)) that are in clinical or preclinical stages. In this review, we focus on emerging strategies other than on ADCs that are either aimed at directly target the HER2 receptor (i.e., novel tyrosine kinase inhibitors) or subsequent intracellular signaling (e.g., PI3K/AKT/mTOR, CDK4/6 inhibitors, etc.), as well as on innovative approaches designed to attack other potential tumor weaknesses (such as immunotherapy, autophagy blockade, or targeting of other genes within the HER2 amplicon). Moreover, relevant technical advances such as anti-HER2 nanotherapies and immunotoxins are also discussed. In brief, this review summarizes the impact of novel therapeutic approaches on current and future clinical management of aggressive HER2 breast tumors.

## 1. Introduction

HER2-positive breast cancers (HER2 BC), which account for about 15–20% of all breast tumors, are highly aggressive neoplasms with poor prognosis [1,2]. These tumors overexpress, due to gene amplification, the oncogenic Epidermal Growth Factor receptor-2 (ErbB2, also known as HER2), which mediates the activation of several intracellular pathways, including MAPK and/or PIK/AKT/mTOR, involved in angiogenesis, invasiveness, metabolic programming, and other processes [3]. HER2 plays a pivotal role in tumor biology, not only in breast cancers but also in other tumors, and, therefore, assessing its overexpression/amplification status is one of the gold standards in the pathological classification of breast carcinomas. In this sense, following the current ASCO/CAP guidelines [4], standardized immunohistochemical HER2 assessment classifies tumors into 0–3+ scores. Tumors with 0–1+ score (≤10% HER2 expression in tumor cells) are classified as HER2-negative, while 3+ (>10% of intense circumferential membrane staining) are considered positive. Additional study is necessary for 2+ HER2-expression tumors (weak-to-moderate complete membrane staining in 10% of tumor cells). In these cases, an equivocal result of HER2 gene amplification should be confirmed by ISH (in situ hybridization), where it is considered positive when the HER2/CEP17 (centromere chromosome 17) ratio is ≥2.0 and the HER2 copy number signals/cell is ≥4 [4].

From the clinical point of view, HER2 typification is an example of breast cancer precision oncology and a successful therapeutic target, since many anti-HER2 targeted treatments have been specifically developed. So far, current targeted HER2 therapies include (i) monoclonal antibodies (MAbs) such as trastuzumab (Herceptin), which was the first approved treatment by FDA (1998), and pertuzumab (Perjeta); (ii) tyrosine kinase inhibitors (TKIs) such as lapatinib, neratinib, and tucatinib; and (iii) antibody–drug conjugates such as ado–trastuzumab emtansine (T–DM1) or trastuzumab–deruxtecan (T–DXd) [5,6,7]. These therapies target different HER2 regions: MAbs bind to the extracellular HER2 region, mainly inhibiting receptor dimerization; TKIs attach to the intracellular tyrosine kinase domain, impeding the HER2-mediated intracellular pathways [5,6,7].

Accordingly, the first-line treatment in the case of neoadjuvant regimens in stage II/III breast cancer patients is currently based on the combination of MAbs (pertuzumab and trastuzumab) and chemotherapy (usually a taxane). The adjuvant treatment setting depends on whether the patient received neoadjuvant therapy. After neoadjuvant treatment, for those patients with residual disease, the recommendation is T-DM1 treatment in the second line, while patients who show a pathological complete response (pCR) receive trastuzumab plus pertuzumab for a year, and then, if necessary, the tyrosine kinase inhibitor neratinib can be applied [8,9]. In addition, patients with breast tumors smaller than 2 cm and who are lymph node negative receive trastuzumab and a taxane, and those with tumors larger than 2 cm or who are lymph node positive (not candidates for neoadjuvant) are treated with chemotherapy, trastuzumab, and pertuzumab [10]. Finally, CLEOPATRA phase III trial results showed the benefit of treatment with pertuzumab plus trastuzumab and docetaxel as the standard-of-care for metastatic patients who have not received any previous anti-HER2 treatment [11] (Figure 1).

Despite the benefits of anti-HER2 therapies in the survival rate of HER2 BC patients [3], unfortunately, some patients do not respond from the beginning, or they develop acquired resistance during the treatment [12]. Indeed, an ever-growing list of resistance mechanisms to standard anti-HER2 therapies has been reported (Table 1). Therefore, the development of novel treatment strategies to overcome these therapeutic failures continues to be a priority in current oncology. To achieve the highest survival rate of HER2 BC patients, in particular those in advanced/metastatic stages, it is necessary to understand and correctly classify the resistance mechanisms to conventional anti-HER2 drugs as well as to develop more effective and less toxic therapeutic approaches. All these aspects are reviewed in the next sections discussing the applicability of diverse novel strategies that are currently under validation in either clinical or preclinical settings. To this end, we performed a comprehensive review of the relevant literature published from 2010 until September 2022 by searching in the PubMed database using the following keywords: “HER2 Breast Cancer” AND “Treatment/drug resistance” OR “Novel/new therapy” OR “Clinical trial” OR “mechanism of resistance”.

## 2. Strategies Already in the Clinic or under Clinical Trials

### 2.1. Development of New HER2-Targeted Drugs

HER2-targeted agents differ in their composition and mechanism of action and can be broadly classified into the following categories: anti-HER2 and pan-HER antibodies (targeting various HER family receptors, such as Sym013), ADCs, antibodies fused to cytotoxic peptides, bispecific antibodies, immunotherapeutic approaches (including peptide vaccines and T-CARs), and TKIs, among others (reviewed in [58]). The importance of ADCs and HER2-directed immunotherapy agents in current and future treatments have been specifically reviewed [14,59]. In this review, we focus mostly on other approaches.

#### 2.1.1. Tyrosine Kinase Inhibitors (TKIs)

Lapatinib (GW572016, Tyverb, GlaxoSmithKline), a dual HER1–HER2 reversible inhibitor, was the first TKI approved for metastatic HER2 BC and is still the only one recommended in combination with endocrine therapy (letrozole) and/or chemotherapy (capecitabine) for hormone receptor-positive (HR+)/HER2+ advanced or metastatic tumors [60]. However, due to its toxicity and the large number of resistance mechanisms documented (Table 1), this TKI is progressively being replaced by novel TKIs. These TKIs differ in their selectivity among HER receptors, binding strength, efficacy, and toxicity profiles (reviewed in [7,61]).

Neratinib (HKI-272; Nerlynx, Puma Biotechnology) is an oral pan-HER TKI that binds irreversibly to HER1, HER2, and HER4. The potent and extensive effect on diverse HER receptors might explain its frequent intestinal toxicity (mostly diarrhea). This agent is capable of overcoming trastuzumab resistance (either innate or acquired) in HER2 BC models [62]. In 2017, based on the results of ExteNET [63], the FDA approved neratinib treatment as extended adjuvant therapy for early-stage HER2 BC patients who have received at least one year of trastuzumab. In this trial, HR+ tumors responded better to neratinib than HR-negative ones. For metastatic patients previously treated with HER2-targeted drugs, the FDA approved neratinib plus capecitabine in 2020 [64]. Interestingly, neratinib shows good CNS penetration and promising therapeutic activity in patients with brain metastases, despite increased diarrhea as a side effect, compared to lapatinib [64,65]. In this setting, neratinib as monotherapy could also provide some therapeutic benefit [66]. However, it shows more efficacy in combination with capecitabine [65], and indeed, current clinical trials are also evaluating its utility in combination with other therapies: fulvestrant (NCT03289039 and NCT01670877, only in HER2-mutant patients) or T-DM1 (NCT02236000 or NCT01494662 in patients with brain metastasis) [7,67].

Tucatinib (ARRY-380, ONT-380, Tukysa, Seattle genetics) is an oral, reversible inhibitor that is >1000-fold more specific for HER2 than for EGFR, and thus shows reduced EGFR-associated toxicity. Preclinical studies showed strong antitumor activity in HER2-overexpressing cancers as a single agent and combined with trastuzumab [68,69,70], including those with truncating p95/p110 mutations, known to cause trastuzumab resistance [71]. Interestingly, tucatinib shows increased activity and central nervous system (CNS) penetration compared to lapatinib or neratinib in mice with intracranial HER2 cancers [72]. Moreover, it produces tumor regression when co-administered with either CDK4/6 inhibitors or hormonal therapy in xenografts models of HER2 cancer [69]. In the pivotal HER2CLIMB trial, patients with metastatic disease and previous anti-HER2 treatment who were given trastuzumab + capecitabine and tucatinib showed a significant increase in PFS and OS compared to the placebo arm; this was also seen in patients with brain metastasis [73]. In April 2020, the FDA approved the triple combination of tucatinib + trastuzumab + capecitabine for patients with advanced disease (including CNS metastasis) that had progressed from previous anti-HER2 treatments [74]. Ongoing clinical trials in different phases are investigating the utility of tucatinib in various drug combinations, such as chemotherapy, T-DM1, T-DXd, the aromatase inhibitor (letrozole), CDK4/6 inhibitors (palbociclib), or immunotherapy (Pembrolizumab) (revised in [67]).

Pyrotinib (SHR1528, Irene) is an irreversible pan-HER (HER1, 2, 4) inhibitor. Preclinical data showed HER2 cancer growth inhibition [75] and a synergistic effect with palbociclib in xenografts [76]. In 2020, it was approved by the Chinese State drug administration in combination with capecitabine as the second line of treatment for patients with advanced or metastatic cancer. In this setting, pyrotinib increased PFS compared to lapatinib, though pyrotinib showed higher rates of toxicity [77]. Multiple clinical trials are ongoing in China in metastatic patients with diverse drug combinations, including chemotherapies and other anti-HER2 agents. Some of the studies are designed to assess pyrotinib utility in patients who have progressed from previous HER2 therapy, where its efficacy in overcoming resistance to standard HER2 treatments is being evaluated [67].

Other irreversible pan-HER inhibitors under clinical evaluation are Canertinib (CI-1033), Poziotinib [78], epertinib (S-22611) [79], varlitinib (ASLAN001; NCT02396108), and TAS0728, which is effective in preclinical models of acquired resistance to HER2-targeted antibodies [80]. However, toxicity concerns are halting further studies on poziotinib [78].

While TKIs can have advantages over antibodies (oral administration, capability to reach CNS metastasis, and multiple HER family targets), they still have important limitations. They generally are not used as single agents, and their gastrointestinal and cardiac toxicity raises safety concerns when combined with other anti-HER2 approaches or chemotherapy. For instance, the combination of either pyrotinib or neratinib with capecitabine (compared to lapatinib) produces worrying rates of grade 3 diarrhea according to PHOEBE [77] and NALA trials [64]; thus, caution should be taken with these TKI + capecitabine combinations. Moreover, cross-resistance to diverse TKIs has been reported [81], and HER2 L755S mutation, although infrequent in breast cancer, can make tumors insensitive to lapatinib, neratinib, and tucatinib [82]. Fortunately, breast cancer cell lines with this mutation seem to be sensitive to either T-DM1, T-DXd [83], or poziotinib [84]. Similarly, in metastatic non-small-cell lung cancer patients with HER2 mutations, T-DM1 [85] and TDXd [86] produced promising anticancer activity, thus providing a new potential way to treat tumors with TKI-resistance mutations.

#### 2.1.2. Novel Antibody-Based Therapies

In contrast to TKIs, therapeutic antibodies can effectively kill cancer cells in vivo through activation of antibody-dependent cellular cytotoxicity (ADCC) by effector lymphocytes. The development of novel antibody-based therapies is leading to a significant revolution in current and future HER2 BC patients, including those refractory to standard therapies. Margetuximab–cmkb (MGAH22, Margenza) is a novel chimeric antibody that binds to HER2 on the same epitope as trastuzumab but contains a constant FC region with a higher affinity for CD16a; thus, this antibody enhances ADCC via NK cells expressing CD16a receptor [87]. Mostly based on data from the SOPHIA trial (NCT02492711), in 2020, the FDA approved its use in combination with chemotherapy for the treatment of metastatic HER2 BC patients who have received at least two previous anti-HER2 therapies [88]. Other novel HER2-specific (e.g., 19H6-Hu) or pan-HER antibody mixtures (such as Sym013, which contains six humanized monoclonal antibodies targeting different epitopes of HER1, 2, and 3 receptors) are currently in preclinical or clinical studies [89,90]. Moreover, bispecific antibodies that bind either to various HER2 epitopes (such as ZW25 and azymetric) or HER2 plus immune receptors (e.g., PRS-343 targeting CD137) have shown promising preliminary results in trials [91,92].

Among antibody-derived approaches, ADCs are already a reality and have resulted in encouraging improvement in patient survival; these include T-DM1 (Kadcyla),T-deruxtecan (Enhertu) and Disitamab vedotin (RC48; Aidixi, approved in China for gastric and urothelial carcinoma), which are already used in the clinic, while a large list of other ADCs, including Trastuzumab Duocarmazine (SYD985), ALT-P7, ARX788, A166, BAT8001, and XMT-1522, among others, are under clinical evaluation (revised in [14,93]). Rather than attaching chemotherapeutic or cytotoxic agents to anti-HER2 antibodies (as ADCs do), immunotoxins contain an anti-HER2 antibody or its fragments fused with cell-death-inducing peptides. These peptides can originate from natural toxins (such as Pseudomonas exotoxin A “ETA”) or mammalian/human anti-tumor proteins (including granzyme B, caspases, endostatin, apoptosis-inducing factor, and RNases, among others) (reviewed in [94]). Many of these fusion proteins can selectively kill HER2 BC cells in vitro and/or in vivo using preclinical models [94]. However, only a few HER2-immunotoxins have so far been evaluated in clinical trials, where they just reached phase I with HER2 cancer patients. Specifically, ScFv (FRP5)–ETA [95] and MT–5111 (containing Shiga-like Toxin A subunit [96]) were generally well-tolerated, but limited efficacy was observed, and erb–38 rIT (using ETA) caused general hepatoxicity with low therapeutic benefit [97]. Therefore, immunotoxins require further refinement to reduce side effects, improve penetration into tumor tissues, and avoid neutralization by the immune system [94].

Despite encouraging clinical benefits observed by recent FDA-approved antibody-based approaches and those still under clinical trials, some resistance mechanisms to these new agents have already been reported. Fortunately, some of them could be potentially reversed by targeting other signaling pathways or by combination with other anti-HER2 drugs or treatments (revised in [14,93]).

#### 2.1.3. Combination Therapy

As commented above, one effective way to overcome resistance to anti-HER2 single agents is the combination of diverse anti-HER2 drugs (antibodies, ADCs, and TKIs) with/without chemotherapy. This is a possibility in current practice (for example, combining tucatinib plus trastuzumab + capecitabine is approved for metastatic disease), and numerous clinical trials are evaluating the safety and effectiveness of diverse combinatorial strategies: for instance, tucatinib + TDM1 or tucatinib + T-DXd (HERCLIMB02-04 trials) (revised in [67]).

Importantly, the focus of many preclinical studies and clinical trials is to demonstrate the benefits of combining anti-HER2 agents with drugs/treatments targeting downstream signaling pathways or common molecular resistance mechanisms (described in the section below). While these combinations could be very effective, they should be designed carefully to avoid additive toxicities. For instance, the NTC03846583 clinical trial proposing the combination of tucatinib + trastuzumab and abemaciclib was withdrawn since metabolic interference could increase the plasma concentration and toxicity of both abemaciclib and tucatinib [98].

#### 2.1.4. Immunotherapies

The therapeutic activation of the immune system to elicit tumor recognition and cancer cell killing is one of the greatest advances in current oncology. For HER2 BC, there are multiple and complementary approaches under pre-clinical or clinical evaluation. Extensive and updated reviews on the importance of different immunotherapies for HER2 BC treatment have been recently published elsewhere (see, for instance [99,100,101]); thus, only the most relevant results are briefly mentioned here. Techniques that use the HER2 molecule as the target for recognition by immune cells, such as HER2 vaccines, bispecific antibodies, and HER2-targeted chimeric antigen receptor T-cell therapy (CAR-T), are very promising, but none of them have been clinically approved so far [99,100]. Regarding HER2 CAR-Ts, this approach can effectively kill trastuzumab-resistant BC tumors in vivo [102]. Currently ongoing phase I clinical trials with HER2-CAR-T in patients with advanced HER2 BC include NCT369630 (including patients with CNS metastasis), NTC04511871 (using autologous T cells), and NCT04650451 (using the dual switch CAR-T BPX-603) [100]. The results of these trials will determine the safety of these approaches, as potentially lethal effects have been described with high doses of HER2-CAR-T in preclinical mouse in vivo studies [103]. Moreover, some mechanisms of resistance have already been described, such as disruption of IFN-gamma signaling, which is a general mechanism for cancer cells to elude immune killing produced by CAR-T cells or bispecific antibodies [104].

With respect to therapeutic vaccines, different technical approaches are used to induce HER2-mediated immunization, including HER2 peptides and DNA-based, viral-vector-mediated, and cell-based vaccines [100]. Vaccines based on E75 or GP2 peptides are currently being tested in multiple clinical trials, with very variable results depending on the study [99,105]. In a recent meta-analysis evaluating 24 clinical trials, You et al. [99] concluded that E75-based vaccines are generally safe, elicit strong immune response, and may provide significant, albeit reduced, clinical benefits. However, clinical trials testing distinct peptide vaccine formulations in combination with either trastuzumab or checkpoint inhibitors are underway [99,105]. Moreover, in addition to metastatic patients, peptide vaccines have been tested in the adjuvant setting to prevent progression to metastatic disease [99,100,105].

Rather than attacking the HER2 molecule, current immunotherapies aim to hamper immune checkpoint inhibitors, and these approaches have been successful in the treatment of triple negative breast cancer (TNBC) [106]. Whereas HER2 carcinomas are usually more inflamed than luminal A tumors [107], the use of immunotherapies in these tumors is still a controversial issue [108]. Four immune checkpoint antibodies are undergoing clinical trials, alone or in combination with other agents: Atezolizumab (binds to PD-L1 and produces dual blockage of PD-1 and CD80 receptors [109]); Pembrolizumab (anti-PD1 antibody that blocks interaction with PD-L1 and PD-L2 ligands [110]); durvalamab (anti-PD-L1 that inhibits interaction with PD-1 and CD80 receptors [111]); and avelumab (anti-PD-L1 [112]). Initial studies in the advanced setting such as PANACEA (Pembrolizumab + Trastuzumab) or KATE-2 (T-DM1 plus Atezolizumab or placebo), showed some clinical response of the immunotherapies only in PD-L1-positive tumors [113,114]. Multiple clinical trials are underway in previously treated advanced/metastatic patients, but since these patients generally have weakened immune systems, other studies are testing the effect of immune checkpoint inhibitors in the adjuvant setting (such as the Astefania trial) or the neoadjuvant setting (revised in [100]). The comprehensive review by Kyriazoglou and colleagues [101] summarizes the results of the clinical trials using these drugs in HER2 BC through 2020, and the authors conclude that checkpoint inhibitors are tolerable and safe, and, overall, show some positive results, but the clinical benefit is still debatable. It is required to optimize adequate biomarkers that can help with proper patient selection, such as tumor PD-L1 status, mutational burden (TMB), or the percentage of tumor-infiltrating lymphocytes (TILs) [101]. Further studies will clarify if these approaches could be useful in patients resistant to previous HER2-targeted drugs.

### 2.2. Therapies Not Directly Targeting HER2 Receptor

#### 2.2.1. CDK4/6 Inhibitors

In HER2 BC, the cyclin D1-CDK4/6-pRb pathway is controlled by cyclin D1 overexpression (*CCND1*) or increased stability, leading to resistance to anti-HER2 therapies [115]. In this sense, preclinical studies revealed that resistant tumor cells exhibited augmented levels of cyclin D1 and CDK4, and the subsequent addition of a CDK4/6 inhibitor restored cell sensitivity to HER2-targeted therapies [115]. These data reinforced previous preclinical studies that pointed out the synergy between CDK4/6 inhibitors and anti-HER2 drugs [116,117].

Currently, three CDK4/6 inhibitors (abemaciclib, palbociclib, and ribociclib) have been approved by the FDA, but are in ongoing development for advanced HER2 BC [118,119]. In this regard, early clinical studies also propose the combination of CDK4/6 inhibitors plus anti-HER2 therapies, particularly in the subset of HER2 BC patients with estrogen receptor (ER)-positive expression [120,121,122]. At present, many clinical trials are evaluating the use of CDK4/6 inhibitors in advanced HER2 BC, highlighting three randomized trials: the PATINA study (NCT02947685) focuses on the benefit of the addition of palbociclib to trastuzumab/pertuzumab + endocrine therapy in HER2+ /HR+ invasive BC pretreated with chemotherapy plus an anti-HER2 therapy; the SOLTI-1303 PATRICIA trial (NCT02448420) studies the combination of palbociclib plus trastuzumab +/− letrozole in pretreated (2–4 prior lines of anti-HER2 drugs) advanced HER2 BC; and the MonarcHER study (NCT02675231) assesses the combination of abemaciclib plus trastuzumab +/− fulvestrant in HER2+ /HR+ locally advanced or metastatic BC pretreated with taxane plus two anti-HER2-based regimens. Initial results from the SOLTI-1303 PATRICIA and the MonarcHER2 trials have already been published, noting in the first case the need to previously classify BC by PAM50 because the PFS after treatment with palbociclib + trastuzumab was significantly higher in luminal vs. nonluminal BC [123]. In the case of the MonarcHER2 trial, the combination of abemaciclib with trastuzumab and fulvestrant revealed significant improvement to the PFS in comparison with standard-of-care chemotherapy [124]. However, the absence of a fulvestrant + trastuzumab arm in this trial limits the interpretation of the benefit of adding abemaciclib. Further, another trial was carried out (NCT02308020), in which treatment with abemaciclib alone or with trastuzumab was evaluated in HR+ /HER2+ BC patients with brain metastases, but no objective responses were observed [125]. Therefore, these data point out the clinical benefit of CDK4/6 inhibitors plus anti-HER2 therapies to overcome resistance in some cohorts, suggesting the need for further clinical trials to introduce CDK4/6 inhibitors in the therapeutic strategy of advanced HER2/HR BC. Indeed, many other nonrandomized trials are in current development: NCT03054363, NCT03530696, and NCT03709082.

#### 2.2.2. PI3K/AKT/mTOR Inhibitors

The PI3K/AKT/mTOR pathway is frequently dysregulated in many cancer subtypes, including BC. Indeed, targeting this axis is an attractive approach to overcome therapy resistance in advanced breast cancer [126,127]. *PIK3CA*-activating or *PTEN*-inactivating mutations, which are two of the most frequent genetic alterations in breast cancer [128,129], lead to abnormal PI3K pathway activation and subsequently increase phosphatidylinositol 3,4,5-triphosphate (PIP3) levels [126]. PIP3 can interact with phosphoinositide-dependent kinase-1 (PDK1) to phosphorylate AKT. Furthermore, AKT may also be phosphorylated by mTORC2. Activated AKT acts as a key regulator, with many downstream effectors and substrates, including mTORC1, different transcription factors such as FoxO, GSK3 and NF-κB, MDM2, and BAD to promote the survival and tumorigenesis of the cell [126,127]. Particularly in HER2 BC, activation of the PI3K/AKT/mTOR pathway by genetic alterations is associated with resistance to anti-HER2 targeted therapies [122]. Indeed, preclinical studies have shown that *PIK3CA* or *PTEN* mutations have a negative result on trastuzumab’s therapeutic effect [23,24], and these data were subsequently confirmed by early clinical trials [130]. Moreover, PI3K/AKT/mTOR pathway hyperactivation has been identified as one of the main reasons for the oncogenicity associated with different mechanisms of resistance to anti-HER2 therapies, such as HER2 truncated mutant (p95-HER2) [131] or HER3 [132] overexpression. Therefore, many inhibitors that target different steps of this pathway (PI3K, AKT, or mTOR inhibitors) have been developed in recent years, with outstanding results in overcoming resistance to anti-HER2 drugs in preclinical studies [133,134,135,136]. Among them, the FDA approved alpelisib, a PI3K inhibitor, and everolimus, an mTOR inhibitor for the treatment of advanced HR+ /HER2− BC, but they have not yet been approved for advanced HER2+ BC.

In this sense, an important randomized phase III trial, BOLERO-3, has demonstrated the PFS benefit of the addition of everolimus to trastuzumab plus vinorelbine in HER2 locally advanced or metastatic BC resistant to trastuzumab (NCT01007942) [137]. Furthermore, data from BOLERO-3 and the randomized phase III BOLERO-1, which evaluated the use of everolimus as the first line in HER2 advanced BC (NCT00876395), showed that everolimus addition is especially effective in patients harboring *PIK3CA* mutations, loss of *PTEN*, or hyperactivation of the PI3K-AKT-mTOR pathway, supporting the relevance of including the status of these markers in BC diagnosis [138]. Other clinical trials to assess the use of everolimus or other mTOR inhibitors (ridaforolimus, temsirolimus, or TAK-228) have been carried out but center on HER2 metastatic BC resistant to trastuzumab (e.g., NCT01783756 [139], NCT00736970 [140], and NCT01111825). Moreover, currently, several clinical trials focusing on treatment with PI3K or AKT inhibitors in HER2 advanced BC resistant to therapy are ongoing. Among these, two trials stand out (NCT04208178 and ALPHABET, and NCT05063786) for studying the use of alpelisib (PI3K inhibitor) in HER2 advanced BC with *PIK3CA* mutations and resistance to trastuzumab and pertuzumab or to T-DM1, respectively.

Interestingly, inhibition of ataxia telangiectasia and Rad3-related kinase (ATR), a member of the phosphatidylinositol 3-kinase-related kinase (PIKKs) family and involved in DNA repair in response to DNA damage and replication stress [141,142], is currently being tested in combination with trastuzumab deruxtecan in HER2 advanced or metastatic BC pretreated with chemotherapy and anti-HER2 therapy in a phase I/Ib trial (DASH, NCT04704661).

#### 2.2.3. Endocrine Therapy

There is complex bidirectional crosstalk between HER2 and the signaling pathways activated by hormonal receptors for estrogen (ER), progesterone (PgR), and androgens (AR) that can lead to cancer resistance to either antihormonal drugs [143] or anti-HER2 agents, and thus, co-targeting HER2 and HRs could lead to tumor regression [144]. Around 50% of HER2 tumors express ER, and most of these patients are treated with anti-HER2 antibodies (usually up to one year), whereas endocrine treatment requires at least 5 years [143]. It is interesting to note that around 20% of tumors negative for HER2 can become positive during progression, mostly after endocrine treatment [145,146]. For early-stage HR/HER2 tumors, neratinib could be used for extended adjuvant therapy, while in metastatic ER/HER2 tumors, the current guidelines recommend chemotherapy and anti-HER2 agents, either TKIs (lapatinib, neratinib, tucatinib, or pyrotinib), monoclonal antibodies (trastuzumab, pertuzumab, or margetuximab), or ADC (T-DM1 or T-DXd), regardless of ER status [143]. Among TKIs, lapatinib is the only one approved in combination with letrozole. While different clinical studies have proven the safety of combining endocrine therapy with anti-HER2 agents in advanced or metastatic settings, the relative benefit of endocrine treatment is still uncertain [143]. However, in these patients, the addition of CDK4/6 inhibitors seems to have clinical benefit in diverse trials [123,124].

Regarding androgens, AR is a potential marker for predicting pathological response in HER2 BC patients treated with trastuzumab plus pertuzumab neoadjuvant therapy [147], and pre-clinical studies with the AR antagonist enzalutamide proved that this agent can avoid trastuzumab resistance [148]. Consequently, phase II clinical trials showed that the combination of enzalutamide with trastuzumab in patients with advanced BC previously treated with trastuzumab was tolerable, and some patients had durable disease control [149].

#### 2.2.4. Hsp90 Inhibitors

Heat shock protein 90 (HSP90) is an essential chaperone that enhances conformational maturation, stability, and activation of oncoproteins, including HER2 [150,151]. In addition, AKT and other members of the HER2 signaling pathway are also clients of HSP90 [152,153,154]. Therefore, HSP90 has been proposed as a potential therapeutic target to induce the degradation of HER2, and preclinical studies have confirmed the beneficial effect of HSP90 inhibitors, even in trastuzumab-resistant cancer cells [155,156,157,158]. Therefore, combinations of different HSP90 inhibitors (e.g., tanespimycin, AUY922, ganetespib, and alvespimycin) with anti-HER2-based drugs have been tested in clinical trials with HER2 BC patients (in different disease stages, including tumors refractory to anti-HER2 drugs) [159,160,161,162]. Unfortunately, despite promising efficacy in preclinical studies, so far none of the HSP90 inhibitors have obtained regulatory approval due to their high off-target toxicity and inadequate pharmacokinetic profiles [163]. Nonetheless, research on novel anti-HSP90 drugs continues and is now focusing on isoform-selective or C-terminal Hsp90 inhibitors that present limited cytotoxic side effects, such as NCT-547, which produces effective degradation of HER2 and the p95HER2 variant, leading to in vivo growth reduction of trastuzumab-resistant tumors [164].

Histone deacetylase (HDAC) inhibitors have also been assessed to overcome therapy resistance in BC [165]. Indeed, HDAC blocking results in HSP90 acetylation, causing dissociation of HSP90 from HER2, with subsequent HER2 degradation [165,166]; also, its combination with anti-HER2 drugs promotes apoptosis by FOXO3-mediated Bim1 expression [167]. Two clinical trials with HDAC inhibitors (entinostat or panobinostat) in combination with anti-HER2 drugs were performed, but results showed modest or no clinical benefit (NCT01434303 [168] and NCT00788931).

#### 2.2.5. Other Receptor Tyrosine Kinase (RTKs) Inhibitors

Several receptor tyrosine kinases (RTKs) have been involved in resistance to anti-HER2 therapies by bypassing HER2 inhibition [169,170]. Hence, insulin-like growth factor 1 receptor (IGF1R) inhibition by cixutumumab has been tested in combination with lapatinib plus capecitabine in locally advanced or metastatic HER2 BC resistant to trastuzumab, but with poor results (NCCTG N0733 and NCT00684983 [171]). Moreover, the multitarget tyrosine kinase inhibitor pazopanib, which blocks vascular endothelial growth factor receptor (VEGFR), platelet-derived growth factor receptor (PDGFR), and c-KIT, in combination with lapatinib, has also been proven in HER2 invasive BC pretreated with chemotherapy and trastuzumab, with a worse or similar PFS (Progression-free survival) compared to lapatinib treatment alone (lapatinib plus placebo) (NCT00558103).

#### 2.2.6. Non-Receptor Tyrosine Kinase Inhibitors

Src is a membrane non-receptor tyrosine kinase that is activated by several receptor tyrosine kinases (RTKs), including HER2, leading to the regulation of the Ras, focal adhesion kinase (FAK), and PI3K pathways and, therefore, enhancing the malignant behavior of the cell [172,173]. In this sense, Src hyperactivation has also been related to limited response to trastuzumab [173,174]. Consequently, several preclinical studies have shown the efficacy of Src targeting by specific inhibitors in combination with anti-HER2 agents [174]. In terms of clinical trials, the Src inhibitor dasatinib, approved by the FDA for leukemia, has been assessed as first line in combination with trastuzumab plus paclitaxel in HER2 metastatic BC (NCT01306942) with awaited results [175], and as second line alone in HER2 recurrent, locally advanced, or metastatic BC pretreated with chemotherapy (taxane and/or anthracycline, NCT00371345) with poor results.

Bruton’s tyrosine kinase (BTK) is a non-receptor tyrosine kinase that plays a vital role in the development and maturation of B cells [176]. Recently, an alternative isoform of BTK (BTK-C) was found to be overexpressed in breast cancer cells and was associated with cell survival [177]. Subsequently, preclinical studies have revealed that BTK inhibition by ibrutinib overcomes resistance to lapatinib in HER2 BC by avoiding activation of the AKT signaling pathway by NRG or EGF [178]. A phase I/II trial testing the combination of ibrutinib plus trastuzumab in T-DM1 pretreated HER2 metastatic BC is currently ongoing (NCT03379428).

#### 2.2.7. FASN Inhibitors

Fatty acid synthase (FASN) regulates the expression, activity, and cellular location of HER2 [179]. In fact, FASN inhibition suppresses HER2 expression by upregulating the transcription factor PEA3, a repressor of HER2 [179,180]. Further, preclinical studies have demonstrated the efficacy of targeting FASN to revert the resistance to anti-HER2 based regimens in HER2 BC [181,182]. This research has paved the way for the use of FASN inhibitors such as TVB-2640 in clinical trials in combination with trastuzumab in HER2 metastatic BC pretreated with anti-HER2 drugs (NCT03179904, currently ongoing).

#### 2.2.8. PARP Inhibitors

Poly (ADP-Ribose) polymerase (PARP) is pivotal for the base excision repair pathway [183]. PARP inhibition has been shown to be a promising therapeutic strategy in homologous recombination repair-deficient tumors, such as BRCA1 and BRCA2-mutated breast cancer [183]. Indeed, the FDA approved several PARP inhibitors in recent years to treat HER2-negative, BRCA1 and BRCA2-mutated BC [183]. However, recent preclinical research has pointed out the effectiveness of using PARP inhibitors in trastuzumab-resistant HER2 BC via inhibition of NF-κB signaling [184]. In this regard, a phase Ib/II trial is being performed, assessing the combination of the PARP inhibitor niraparib plus trastuzumab in HER2 metastatic BC pretreated with anti-HER2 drugs (NCT03368729).

## 3. Strategies in Pre-Clinical Development

Although a wide spectrum of therapeutic strategies has entered clinical trials, preclinical research continues to unravel the molecular mechanisms that underlie anti-HER2 therapy resistance, and this knowledge has led to the development of novel potential therapeutic approaches against other molecular targets. Indeed, protective autophagy and/or the role of other genes located on the HER2 amplicon that are co-amplified/overexpressed with the receptor have emerged as responsible for anti-HER2 therapeutic failure. For instance, insights about novel therapeutic approaches such as nanotherapy or even microbiota have been strategically proposed to overcome this resistance.

### 3.1. Protective Autophagy Blockade

Autophagy is a multistep process whereby cells degrade dysfunctional or unnecessary intracellular components under stress conditions. Further, it may act as a pro-survival (by obtaining energy) or a pro-death (by degrading cellular components) mechanism depending on the cellular context [185]. In HER2 BC, autophagy has been considered a tumor-inhibiting process because of the activation of several oncoproteins involved in HER2 signaling, such as both mTOR and AKT, which provokes autophagy inhibition [185,186]. However, protective autophagy has come out as a novel resistance mechanism to anti-HER2 therapies [41,42]. In this sense, preclinical studies showed that trastuzumab- or lapatinib-refractory HER2 BC cells exhibit an increase in autophagosome formation, which was essential for their survival [41,42]. Therefore, blocking this protective autophagic process might be a suitable approach to restore sensitivity. Indeed, preclinical studies demonstrated that the combination of autophagy inhibitors, such as chloroquine, plus anti-HER2 therapies boosts tumor cell death [41,42,187,188].

Few clinical trials have been performed analyzing the use of chloroquine or hydroxychloroquine in breast cancer, and none of them so far have distinguished between breast cancer subtypes [185]. Due to the importance of the cellular context in autophagy-mediated drug resistance, future clinical trials need to consider the cancer subtype and the previous treatment history. In this regard, new trials are ongoing to evaluate the usage of hydroxychloroquine in HR+ /HER2− BC (NCT03032406, NCT03774472, and NCT04316169) due to autophagy to promote resistance to other drugs [185,186]. However, further research is crucial to decode the mechanisms whereby autophagy promotes resistance to anti-HER2 therapies to test their combination with autophagy inhibitors in clinical trials and to define a patient profile better suited to this therapeutic approach.

### 3.2. Targeting HER2 Amplicon and Neighbor Genes

The consensus “HER2 amplicon” in breast cancers located at the 17q12–21 chromosomal region contains (besides *ERBB2*) at least five other genes (*STARD3*, *TCAP*, *PNMT*, *PGAP3*, and *MIEN1*). Interestingly, in more than 60% of HER2-amplified breast tumors, the amplicon is larger, encompassing up to 27 genes [48]. While HER2 is considered the main oncogene within this amplicon, studies have shown that some ERBB2-amplified breast cancer cell lines are not truly addicted to the HER2 oncogene but require other genes for survival against therapeutic challenge [48,189]. In fact, multiple studies indicate that other genes within the HER2 amplicon or adjacent chromosomal regions play important roles in the pathobiology and the treatment behavior of HER2 tumors, supporting that these genes are not mere “passenger” genes [48]. Specifically, Sahlberg and colleagues proved that silencing either *STARD3*, *GRB7*, *PSMD3*, *PERLD1*, *PPP1R1B*, *THRA*, or *GSDMA* partially sensitized specific breast cancer cell lines to anti-HER2 agents [48]. Thus, identification of the co-amplified genes that affect cancer development and drug response is a suitable approach for the development of new therapeutic strategies in these tumors. In this sense, among 22 genes closely associated with the HER2 amplicon, the overexpression of *GSDMB* (Gasdermin-B; located at 17q12) had the strongest statistical association with poor prognosis in diverse microarray datasets [43]. Further validation using a GSDMB-specific monoclonal antibody and FISH probe confirmed that *GSDMB* amplification and protein overexpression occurs in more than 60% of HER2 BC [43,45]. Importantly, in these tumors, GSDMB overexpression correlates to distant metastasis, poor prognosis (relapse-free survival and pathologic complete response), and reduced response to standard treatment (trastuzumab plus chemotherapy) independent of HR status in both neoadjuvant and adjuvant settings [43]. In fact, GSDMB upregulation is functionally involved in mediating HER2 BC’s aggressive behavior in multiple ways, since it enhances cancer cell migration and invasion, in vivo tumorigenesis (in GEMMs and orthotopic xenografts [44,190]), metastasis [44,45], and resistance (innate and acquired) to anti-HER2 therapies [43,188]. GSDMB induces survival to anti-HER2 agents through the promotion of pro-survival autophagy in breast and gastric HER2 cancers [188]. These pro-tumor functions could be partially reversed by GSDMB silencing via si/shRNAs [44,45,188]. To further prove that GSDMB is a novel and feasible therapeutic target in HER2 BCs, an innovative nanotherapy (nanoparticles loaded with a specific anti-GSDMB antibody) was generated and showed effectiveness in reducing GSDMB pro-tumoral activities, including resistance to therapy (discussed below in Section 3.3) [45].

Paradoxically, GSDMB shares a potentially activatable pro-cell-death function with other members of the Gasdermin family. Indeed, Gasdermins are the key effectors of a lytic and pro-inflammatory cell death mechanism termed pyroptosis, and thus, triggering pyroptosis in tumors has been shown to induce a potent antitumor immune response and cancer regression (revised in [191]). Since the HER2 amplicon frequently contains two GSDM genes (*GSDMA* and *GSDMB*), rather than inhibiting *GSDMA–B* expression, activating their pyroptotic activity via therapeutic intervention (either with GSDM-targeted nanotherapies, chemotherapy, or immunotherapy) could be a promising approach for future treatment of HER2 BC resistant to standard therapies (revised in [191]).

Aside GSDMs, there are other frequently amplified genes near the HER2 amplicon that could affect HER2 BC clinical behavior, such as *TOP2A* (in 17q21.2 [192]) or genes within the 17q23 amplicon such as *RPS6KB1*, *PPM1D*, or *MIR21* [193,194]. *PPM1D* (located at the 17q23 region) encodes the wildtype p53-induced phosphatase1 (WIP1), a protein that negatively regulates p53 function, enhances cell cycle progression, and accelerates tumor incidence in HER2 BC mouse models [195]. In HER2/WIP1-amplified tumors, pharmacological WIP1 inhibition reduced cell proliferation and, importantly, also partially restored sensitivity to trastuzumab [193,194], thus suggesting that the WIP1 oncogene could be another potential therapeutic target to reverse anti-HER2 therapy resistance.

### 3.3. Targeted Nanotherapy

Nanotechnology has come out as an interesting way to reduce toxicity and improve different pharmacokinetic and pharmacodynamic aspects of current treatments. These new strategies are based on nanoparticle delivery systems (nanocarriers) composed of biomaterials/biodegradable materials that enhance the action range by increasing the drug–site contact time [196,197]. In recent years, several non-targeted nanotherapeutics, which consisted of improved chemotherapy-delivery systems (such as DOXIL, MYOCET, NANOXEL, etc.), have been approved by regulatory agencies [196]. However, new problems and challenges such as the lack of target specificity and novel toxicities appeared related to these novel treatments [196]. For this reason, current research is focused on targeted nanocarriers directed against HER2 BC cells.

Multiple designs of targeting nanocarriers for HER2 BC have been developed not only as therapy but also as tumor imaging techniques, with the HER2 BC targeting ligand the main difference between designs. In this regard, antibodies are the most common ligand to target HER2 BC cells, with the employment of trastuzumab in the lead, usually combined with other therapeutical agents [196]. For instance, Lin and colleagues developed a nanocarrier based on a liposome–PEG–PEI complex that was conjugated with trastuzumab plus curcumin or doxorubicin, proving that the nanomedicine increased the therapeutic effect compared to trastuzumab alone both in vitro and in vivo, specifically in HER2 BC cell lines [198]. Interestingly, trastuzumab-conjugated nanocarriers have also been studied in other HER2-positive cancers, such as HER2 gastric cancers, showing promising results in overcoming resistance to trastuzumab [199]. Additionally, photothermal and photodynamic nanotherapy has emerged in the last years, and, particularly in this context, several studies have succeeded in the use of anti-HER2 nanoparticles that accumulated in breast tumors, and after irradiation or illumination, they selectively induced tumor inhibition even in trastuzumab-resistant tumors [200,201,202].

In addition to entire HER2 antibodies, other molecules that have been successfully used as ligands to target HER2 receptor include antibody fragments, nucleic acids, peptides, DARPINS (Designed Ankyrin Repeat Proteins, which are small single-chain scaffold non-immunoglobulin proteins that are less immunogenic than antibodies), or affibodies and ADAPTs (albumin binding domain (ABD)-derived affinity proteins) [196]. Their application as an alternative anti-HER2 therapy or even as a delivery strategy for other therapeutical agents opens the door to a wide variety of treatments to restore sensitivity. Examples of these approaches are DARPin_9-29 (which binds to a HER2 extracellular region different to the pertuzumab epitope) coupled with gold nanorods, which can reduce in vivo HER2 tumor growth [203]; or the ZHER2:2891 affibody conjugated with 5-Fluorouracil, which is effective in killing HER2 cancer cells [204]. Moreover, Truffi et al. conjugated multiple half chains of trastuzumab onto magnetic iron oxide nanoparticles (MNP-HC), achieving an increased effect compared to trastuzumab alone, even in trastuzumab-resistant cells [205]. Further, no resistance to therapy was found after treatment with nanoparticles carrying a siHER2, proving more durable HER2 inhibition than conventional anti-HER2 drugs [206]. Similarly, the use of branched polyethylenimine-functionalized carbon dot (BP-CD) nanoparticles that carry a siHER3 together with trastuzumab showed an improved effect compared to trastuzumab alone [207]. Lastly, the use of anti-HER2 peptides as ligand has allowed, for instance, for improving the efficacy and drug delivery of liposomal doxorubicin [208]. In addition, Xiang et al. developed magnetosome nanoparticles functionalized with an anti-EGFR/HER2 peptide that had promising results for magnetic resonance imaging of EGFR and HER2 tumors [209].

In parallel to anti-HER2 targeted nanomedicines, nanotherapies that specifically attack other molecules important for HER2 tumor behavior have obtained promising preclinical results, even as a novel way to limit resistance to anti-HER2 drugs [45]. In this sense, Molina-Crespo et al. generated biocompatible nanocapsules functionalized with hyaluronic acid that carry an antibody directed against the GSDMB protein (which induces resistance to trastuzumab and lapatinib [45,188]). This targeted nanomedicine (termed AbGB-NCs) increased the sensitivity to trastuzumab in vitro, but also reduced cancer cell migration as well as in vivo breast tumor growth and lung metastasis in orthotopic xenografts, specifically in GSDMB-overexpressing cells, without systemic toxicity [45]. Other novel key molecules that can be targeted by nanotherapy include WIP1 and miRNAs (which otherwise are almost unreachable). Indeed, the specific delivery of miR-21 and WIP1 inhibitors by pH-sensitive nanoparticles has been successfully used to overcome resistance to trastuzumab [194].

Despite the ease of the design and the development of nanocarriers, this technology faces some important challenges in becoming a reality in clinical practice. Among them, the lack of standardization in terms of particle size, toxicity, and surface charge, together with problems with short half-life, stability, and the rate of drug release, suggests that this technology needs further refinement before entering clinical trials [196]. However, the recent approvals by regulatory agencies of lipid nanoparticles conjugated with siRNA (Patisiran) for the treatment of hereditary transthyretin amyloidosis [210] and the two COVID-19 nanoparticle-based vaccines [211] have paved the way for the expansion of this technology in the following years.

### 3.4. Other Approaches

Accumulating evidence shows that resistance to anti-HER2 treatment is a complex process that involves signaling crosstalk with diverse cells within the tumor microenvironment. In fact, macrophages, alveolar epithelial cells, and fibroblastic reticular cells have been shown to act as drivers of both preexisting and acquired resistance to anti-HER2 agents, and thus, microenvironment and tumor heterogeneity should be considered when designing new treatments [212]. The phenotypic plasticity of tumor cells, which is controlled by both intrinsic and microenvironmental cues, can result in the appearance of mesenchymal-like or cancer stem cell-like cells with reduced sensitivity to anti-HER2 drugs [213,214]. Interestingly, the phenotypic evolution of HER2 tumors could lead to the emergence of novel druggable targets, such as PDGFR-B, that can counteract HER2-targeted treatment desensitization [215]. Intrinsic mechanisms such as metabolic rewiring of tumor cells are another source of drug resistance, in particular, lipid metabolism plays an important role in this process. Thus, dysregulation of cholesterol homeostasis using drugs such as lovastatin, in combination with lapatinib, shows an enhanced therapeutic effect over lapatinib alone in HER2 BC patients (revised in [216]). Extrinsic factors such as the gut microbiota can also affect therapy response. In this sense, there is an association between trastuzumab efficacy and gut microbiota due to its modulation of the host immune system. In fact, antibiotic-treated microbiota showed reduced dendritic cell activation as well as IL12p70 release, which is a mechanism needed for trastuzumab efficacy [217]. Therefore, the alteration of the gut microbiota might be a promising strategy to enhance trastuzumab effectiveness or even overcome resistance in the future.

## 4. Conclusions

Treatment options for HER2 BC patients have increased steadily in recent years, and with a myriad of novel approaches under preclinical investigation or clinical validation (Figure 2), it is expected that in a few years, the conventional anti-HER2 antibodies and TKIs will be replaced by more effective and innovative therapies (ADCs, bispecific antibodies, CAR-T cells, nanotherapy, and immunotherapy). However, it is also anticipated that new mechanisms of resistance might also arise to these novel agents; thus, a combination of diverse approaches (including targeted therapies, chemotherapy, hormone therapy, and novel techniques) is likely to be the usual practice in the future of diverse clinical settings. Whereas new agents (such ADCs) are already making a strong impact on overall survival and quality of life of HER2 BC patients in particular scenarios, successfully treating metastatic disease, in particular to the CNS, will be the toughest challenge in the near future.

## Figures and Tables

**Figure 1 cancers-14-04543-f001:**
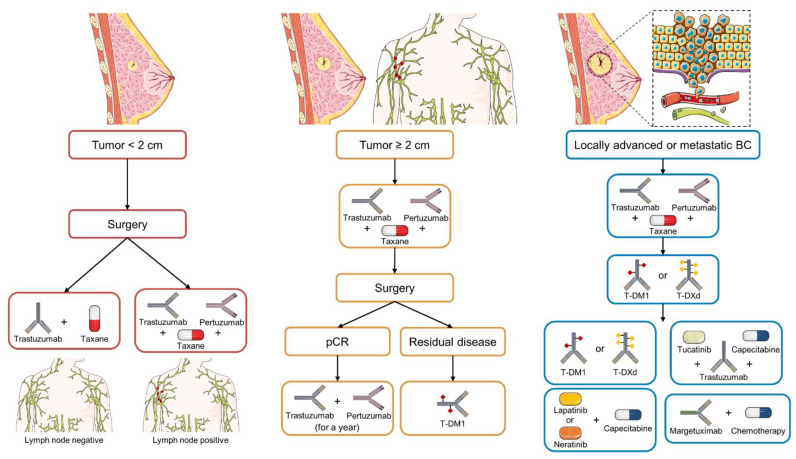
Current established treatment protocol for early-stage, advanced, or metastatic HER2 BC according to ASCO and NCCN guidelines. The different treatment options for each stage are presented sequentially based on their strength of recommendation. Icons designed with smart.servier.com and Adobe Illustrator.

**Figure 2 cancers-14-04543-f002:**
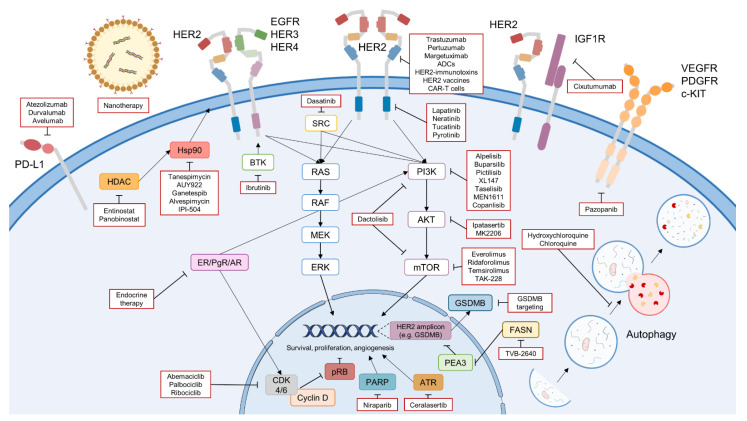
Overview of the novel therapeutic strategies to overcome resistance to anti-HER2 therapies in HER2 BC (highlighted in red). The figure includes the therapies under clinical validation, including novel anti-HER2 drugs (novel TKIs, antibodies, ADCs, CAR-T cells, etc.), immunotherapies, CDK4/6 inhibitors, PI3K/AKT/mTOR inhibitors, endocrine therapy, Hsp90 and HDAC inhibitors, other RTKs inhibitors, non-receptor tyrosine kinase inhibitors, FASN inhibitors, and PARP inhibitors. Furthermore, novel approaches under preclinical validation are also pointed out, such as nanotherapy, inhibition of pro-survival autophagy, and targeting of HER2 amplicon genes (e.g., GSDMB). Icons designed with smart.servier.com and Adobe Illustrator.

**Table 1 cancers-14-04543-t001:** Previously described mechanisms of resistance to anti-HER2 therapies.

HER2-Targeted Therapy	Mechanism of Resistance	References
Trastuzumab/ Pertuzumab/TKIs/T-DM1	HER2 somatic mutations	[13,14,15,16]
Trastuzumab/T-DM1	Expression of truncated receptor variants, such as p95HER2 and Δ16-HER2	[14,17,18]
Trastuzumab/TKIs	Upregulation of ligands and/or other receptors of the HER family (e.g., EGFR, HER3)	[19,20,21,22]
Trastuzumab/TKIs/T-DM1	Overactivation of PI3K/Akt/PTEN/mTOR signaling pathway (via PI3K mutations or loss of PTEN function)	[20,23,24,25,26]
Trastuzumab/Lapatinib	Stimulation of additional RTKs/growth factors (MET tyrosine kinase, insulin-like growth factor 1, VEGF, AXL) and/or downstream kinases (Src) or signaling pathways (MAPK)	[27,28,29,30]
Trastuzumab/TKIs	Inhibition of cell death mechanisms (e.g., overexpression of XIAP or MCL-1)	[31,32,33]
Trastuzumab/TKIs	Alterations in cell cycle regulators (loss of p27Kip1, upregulation of Cyclin E, or activation of CDK12)	[34,35,36]
Trastuzumab	Masking the trastuzumab binding site on HER2 receptor via overexpression of large glycoproteins (MUC4)	[37]
Trastuzumab	Alterations in the Fc receptor (including FCγRIIa polymorphisms) preventing trastuzumab antibody-derived cellular cytotoxicity (ADCC)	[38]
Lapatinib	Crosstalk with endocrine receptor signaling (ER, AR)	[39,40]
Trastuzumab/Lapatinib	Induction of protective autophagy	[41,42]
Trastuzumab/TKIs	Functional involvement of other genes within HER2 amplicon (e.g., *GSDMB*, *STARD3*, *GRB7*, *CDK12*)	[36,43,44,45,46,47,48]
Lapatinib	Phenotypic cell plasticity (epithelial mesenchymal transition and metabolic rewiring)	[49,50]
Trastuzumab/ Pertuzumab/T-DM1/TKIs	Interaction with stromal and immune cells and microenvironmental response to stimuli (chemokines, hypoxia)	[36,51,52,53,54,55]
Trastuzumab	Modulation of specific miRNAs	[56]
Trastuzumab	Up/downregulation of different genes via alterations to transcriptome and chromatin landscape (e.g., *PPP1R1B*)	[57]

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
