# Peer review of "Novel Therapies and Strategies to Overcome Resistance to Anti-HER2-Targeted Drugs"

_cancers, 2022, doi:10.3390/cancers14184543_

Round 1
Reviewer 1 Report
1. Title; suggest you remove “other than ADCs” from the title as you explain it clearly in the abstract. Or ideally, include novel ADCs in development.
2. Introduction; Line 68; there is no role for adjuvant T-DXD outside clinical trials, please remove this as presently only T-DM1 has proven benefit in this setting
3. Methods; please add a section to detail your search criteria for this review
4. Table 1; HER2 mutations can also be a mechanism of resistance to TKIs as you have stated in the final paragraph on page 4, so please add this to the table
5. Strategies already in the clinic or under clinical trials; suggest remove those already in the clinic as these are not novel unless delivered in combination, therefore delete lapatinib and most of the neratinib sections, leaving just the study combining neratinib an ET. This could then allow space for discussion of other neratinib combination studies eg with T-DM1
6. Pyrotinib; this has become a standard of care in China, but the high rates of grade 3 diarrhoea with capecitabine in the PHOEBE trial are similar to that seen with neratinib/capecitabine in NALA, which limits the use and potential novel combinations of both of these drugs.
7. Page 4 final paragraph; please remove the reference to afatinib as development in breast cancer ceased several years ago.
8. Page 4 final paragraph; neratinib has (albeit limited) monotherapy activity (TBCRC 022)
9. Page 4 final paragraph; L755S mutations can still be sensitive to ADCs including TDM1 and T-DXD in lung cancer, so this should be discussed
1. Immunotherapies; please discuss the KATE-2 study results here and the implication for further development in studies such as ASTEFANIA.
1. Immunotherapies; please expand the section on CAR-T cell to include current studies such as NCT03696030
1. Immunotherapies; vaccine studies should be added here
1. CDK4/6 inhibitors; the absence of a fulvestrant/trastuzumab arm limited the interpretation of the Monarch-HER study regarding any additive benefit from abemaciclib, please add this point
1. Page 9, line 398, please change “outstanding” to “awaited” given potential misunderstanding of the former word due to dual meanings
1. Metformin; please add the large negative MA.32 study presented at ASCO 2022 if you keep this section in your manuscript
1. Page 10, line 464, this is very speculative, so I would advise removing this sentence.
1. Page 12, line 539, please qualify what you mean by a better therapeutic effect; you have referenced a review rather than a study and this should be qualified as to whether these hypothesis-generating pre-clinical results are in cell lines, mouse models etc.
1. Please tabulate current clinical trials of interest
1. The figure is an excellent summary. A second figure detailing current standard treatment paradigms for HER2+ breast cancer might be useful for readers to appreciate where novel combinations/drugs might fit in.
Author Response
Rebuttal point by point:
Reviewer 1:
We would like to thank the reviewer for her/his implication in improving the manuscript relevance. The suggested modifications have been highlighted in yellow in the manuscript and commented as follow:
- Title; suggest you remove “other than ADCs” from the title as you explain it clearly in the abstract. Or ideally, include novel ADCs in development.
We fully agree with the reviewer comment, and we have now removed “ADC” from the title. Furthermore, following reviewer suggestion, we now briefly mentioned other ADCs in development (lines 208-210), but we do not expand this information as there is a review specifically focused on ADCs already published for this special journal issue (Diaz-Rodriguez et al 2022; cancers14010154).
- Introduction; Line 68; there is no role for adjuvant T-DXD outside clinical trials, please remove this as presently only T-DM1 has proven benefit in this setting
Duly noted and corrected.
- Methods; please add a section to detail your search criteria for this review
We have included a small paragraph explaining the search criteria at the end of the introduction (lines 92-96). We nevertheless consider that it might be unnecessary to include a separate Methods section, since this is unusual in this kind of review manuscripts.
- Table 1; HER2 mutations can also be a mechanism of resistance to TKIs as you have stated in the final paragraph on page 4, so please add this to the table
This is now clearly stated in Table 1. Additionally, we have now completed this table with further information to summarize the potential resistance mechanisms to other TKIs and anti-HER2 therapies.
- Strategies already in the clinic or under clinical trials; suggest remove those already in the clinic as these are not novel unless delivered in combination, therefore delete lapatinib and most of the neratinib sections, leaving just the study combining neratinib an ET. This could then allow space for discussion of other neratinib combination studies eg with T-DM1
Thanks for this suggestion. However, to offer a general view of the topic to a broader audience (including those readers not expert in the field) we consider that those therapies already in the clinic (like lapatinib and neratinib) should be briefly mentioned. This allows better comprehension of the advantages of the novel drugs or combination applications (under clinical trials). Nonetheless, following reviewer suggestion, we have reduced the section on lapatinib/neratinib current use and have included a short paragraph focused on novel combinations with neratinib under clinical trials (lines 129-135, 138-142). We hope the reviewer find these modifications satisfactory.
- Pyrotinib; this has become a standard of care in China, but the high rates of grade 3 diarrhoea with capecitabine in the PHOEBE trial are similar to that seen with neratinib/capecitabine in NALA, which limits the use and potential novel combinations of both of these drugs.
We have now commented the issue of potential toxic effects of pyrotinib and neratinib in combination treatments (lines 163-166 and 179-182), based on the relevant studies (Saura et al 2020; JCO.20.00147 and Xu et al 2021; S1470-2045(20)30702-).
- Page 4 final paragraph; please remove the reference to afatinib as development in breast cancer ceased several years ago.
The manuscript does not longer mention afatinib and the reference about this drug have been removed (previous reference 68).
- Page 4 final paragraph; neratinib has (albeit limited) monotherapy activity (TBCRC 022)
Taking this point in consideration, we now mention neratinib monotherapy use (line 138). Accordingly, we have corrected the sentence (previous line 161) and no longer states that all TKIs only work in combination with other treatments.
- Page 4 final paragraph; L755S mutations can still be sensitive to ADCs including TDM1 and T-DXD in lung cancer, so this should be discussed.
We have now briefly discussed this matter (lines 184-188) based data from relevant published papers (see references 83-86).
- Immunotherapies; please discuss the KATE-2 study results here and the implication for further development in studies such as ASTEFANIA.
- Immunotherapies; please expand the section on CAR-T cell to include current studies such as NCT03696030
- Immunotherapies; vaccine studies should be added here.
Response to comments 10-12: We acknowledge that immunotherapy approaches will play a key role in the future HER2 BC treatment and deserve more in-depth analyses. However, several extensive and updated reviews and meta-analyses on this topic have been recently published (see for instance refs 99-101 and 105) and therefore, only some of the most relevant results and clinical trial could be briefly mentioned in our manuscript. Following reviewer recommendation, this section (2.1.4) have been expanded, to comment KATE-2 clinical trial results and their subsequent implication (lines 286-292), as well as to include current CART or vaccine studies in HER2 BC (Lines 249-276).
- CDK4/6 inhibitors; the absence of a fulvestrant/trastuzumab arm limited the interpretation of the Monarch-HER study regarding any additive benefit from abemaciclib, please add this point
Now, we have added this point (line 328-329)
- Page 9, line 398, please change “outstanding” to “awaited” given potential misunderstanding of the former word due to dual meanings
We have modified the text as suggested (line 457)
- Metformin; please add the large negative MA.32 study presented at ASCO 2022 if you keep this section in your manuscript
We have eliminated the metformin section since, according to the reviewer comment, data about this treatment is still a controversial issue in BC, and it is not clear its clinical benefit and the next step in metformin applicability for the HER2 BC treatment.
- Page 10, line 464, this is very speculative, so I would advise removing this sentence.
Now, we have removed this specific phrase to improve the manuscript clarity (line 516-518).
- Page 12, line 539, please qualify what you mean by a better therapeutic effect; you have referenced a review rather than a study and this should be qualified as to whether these hypothesis-generating pre-clinical results are in cell lines, mouse models etc.
We have modified this paragraph (line 589-591) to clearly specify the results and methods utilized in that study (Lin et al 2019; s12951-019-0457-3).
- Please tabulate current clinical trials of interest
Thanks for the suggestion. Please note that the main objective of our review was to provide and overall view of several types of future therapeutic approaches in drug-resistant HER2 BC rather than performing an exhaustive compilation of clinical trials. There are excellent specific reviews on each of the anti-HER2 treatment approaches and combinations that include large tables with comprehensive data on clinical trials, such Agostinello et al 2022 (focused on immunotherapy), Kyriazoglou et al 2020 (immunotherapy), Ulrich and Okines 2021 (TKIs), Schlam and Swain 2021 (TKIs), You et al 2021 (vaccines) or Bhoga et al 2022 (CNS metastasis therapies), among others. Therefore, mentioning all relevant clinical trials in all kinds of therapy approaches would generate a very large table.
- The figure is an excellent summary. A second figure detailing current standard treatment paradigms for HER2+ breast cancer might be useful for readers to appreciate where novel combinations/drugs might fit in.
We really appreciate the reviewer comment. Based on his/her suggestion we have now included an additional figure (new Figure 1) summarizing the main current treatment options for HER2 BC according to the disease status. Please note that this figure might not cover all the possible drug combinations and treatment scenarios but the most frequent options.
Reviewer 2 Report
Extensive review of mechanisms of resistance of anti-HER2 targeted treatements.
The authors know well the biology but perhaps less the clinical aspects and treatments:
Line 37 HER2 positive breast cancers
Line 39 These tumor overexpress due to gene amplification (if no gene amplification HER2- according to ASCO guidelines
Line 68 the recommendation is T-DM1 Not T-DXd treatment. Confusion between adjuvant early setting of the disease and metastatic….T-DXd after neo-adjuvant therapy with residual disease is still evaluated in an ongoing clinical trial Destiny Breast 05 whereas in the metastatic setting according to Destiny04 T-DXd is the standard of care as second line after taxane pertuzumab and trastuzumab.
Line 79 metastatic breast cancer
94 pan-HER antibodies?
103 in combination with endocrine therapy (letrozole) or chemotherapy (capecitabine)
139 trastuzumab
144 T-DM1, T-DXd
262 PATINA clinical trial is evaluating a maintenance: addition of palbociclib to letrozole and dual HER2 blockade
266 delete strongly…
308 PFS benefit of the addition of everolimus to trastuzumab plus vinorelbine
326 proved ?
342 Letrozole is not an anti-estrogen : aromatase inhibitor
427 I would be more cautious about metformin. Even if there is a biological rational no strong data to support the development to date. METTEN trial was a very small phase II and the efficacy couldn’t be proven.
Author Response
Rebuttal point by point:
Reviewer 2:
We thank the reviewer for her/his implication in the revision of this manuscript which significantly improve its meaning. The specific comments have been changed and included in the manuscript and they are addressed as follows:
Line 37 HER2 positive breast cancers
Duly noted and corrected.
Line 39 These tumor overexpress due to gene amplification (if no gene amplification HER2- according to ASCO guidelines
Duly noted and corrected.
Line 68 the recommendation is T-DM1 Not T-DXd treatment. Confusion between adjuvant early setting of the disease and metastatic….T-DXd after neo-adjuvant therapy with residual disease is still evaluated in an ongoing clinical trial Destiny Breast 05 whereas in the metastatic setting according to Destiny04 T-DXd is the standard of care as second line after taxane pertuzumab and trastuzumab.
Duly noted and corrected.
Line 79 metastatic breast cancer
We apologize but we do not clearly understand the meaning of this comment. Indeed, to our knowledge and according to all revised information to generate this review, the available and novel HER2 targeted therapies have supposed a clear clinical benefit not only metastatic but also primary tumors.
94 pan-HER antibodies?
We have now clarified the meaning of these approaches and provided an example of them (lines 112 and 201-202).
103 in combination with endocrine therapy (letrozole) or chemotherapy (capecitabine)
Corrected accordingly.
139 trastuzumab
Typo corrected.
144 T-DM1, T-DXd
Corrected accordingly.
262 PATINA clinical trial is evaluating a maintenance: addition of palbociclib to letrozole and dual HER2 blockade
According to the reviewer recommendations we have modified this sentence (lines 315-316)
266 delete strongly…
Duly noted
308 PFS benefit of the addition of everolimus to trastuzumab plus vinorelbine
We have modified this phrase (line 362).
326 proved ?
We have changed this word to “tested” (line 380)
342 Letrozole is not an anti-estrogen : aromatase inhibitor
We apologize this mistake, now we have clarified this specific point (line 156)
427 I would be more cautious about metformin. Even if there is a biological rational no strong data to support the development to date. METTEN trial was a very small phase II and the efficacy couldn’t be proven.
We have eliminated the metformin section since, according to the reviewer comment, data about this treatment is still a controversial issue in BC, and it is not clear its clinical benefit and the next step in metformin applicability for the HER2 BC treatment.